# Isospin competitions and valley polarized correlated insulators in twisted double bilayer graphene

Le Liu[1,2], Shihao Zhang[3], Yanbang Chu[1,2], Cheng Shen[1,2], Yuan Huang[4], Yalong Yuan[1,2], Jinpeng Tian[1,2], Jian Tang[1,2], Yiru Ji[1,2], Rong Yang[1,5], Kenji Watanabe[6], Takashi Taniguchi[7], Dongxia Shi[1,2,5], Jianpeng Liu[3,8], Wei Yang[1,2,5 ✉] & Guangyu Zhang[1,2,5 ✉]

New phase of matter usually emerges when a given symmetry breaks spontaneously, which can involve charge, spin, and valley degree of freedoms. Here, we report an observation of new correlated insulators evolved from spin-polarized states to valley-polarized states in twisted double bilayer graphene (TDBG) driven by the displacement field ($D$). At a high field $|D| > 0.7$ V/nm, we observe valley polarized correlated insulators with a big Zeeman $g$ factor of ~10, both at $v = 2$ in the moiré conduction band and more surprisingly at $v = -2$ in the moiré valence band. Moreover, we observe a valley polarized Chern insulator with $C = 2$ emanating at $v = 2$ in the electron side and a valley polarized Fermi surface around $v = -2$ in the hole side. Our results demonstrate a feasible way to realize isospin control and to obtain new phases of matter in TDBG by the displacement field, and might benefit other twisted or non-twisted multilayer systems.

[1] Beijing National Laboratory for Condensed Matter Physics and Institute of Physics, Chinese Academy of Sciences, Beijing 100190, China. [2] School of Physical Sciences, University of Chinese Academy of Sciences, Beijing 100190, China. [3] School of Physical Sciences and Technology, ShanghaiTech University, Shanghai 200031, China. [4] Advanced Research Institute of Multidisciplinary Science, Beijing Institute of Technology, Beijing 100081, China. [5] Songshan Lake Materials Laboratory, Dongguan 523808, China. [6] Research Center for Functional Materials, National Institute for Materials Science, 1-1 Namiki, Tsukuba 305-0044, Japan. [7] International Center for Materials Nanoarchitectonics, National Institute for Materials Science, 1-1 Namiki, Tsukuba 305-0044, Japan. [8] ShanghaiTech Laboratory for Topological Physics, ShanghaiTech University, Shanghai 200031, China. ✉email: wei.yang@iphy.ac.cn; gyzhang@iphy.ac.cn

Twisted graphene-based moiré system is an ideal platform to discover new quantum phases of matter[1–14]. The formation of narrow moiré flat bands quenches the kinetic energy[15], and induces correlated phases as a result of an enhanced Coulomb interaction with respect to the kinetic energy. Each conduction or valence moiré band has four isospin flavors, involving spin and valley degree of freedoms. The interplay between charge, spin and valley (orbital) yields a rich phase diagram, in which some delicate yet fragile quantum phases are accessible at extreme conditions. The pronounced electron interaction in flat bands tend to induce a spontaneous symmetry breaking at integer fillings[16–18]. For instance, valley polarization helps to unveil the nontrivial topological nature of the moiré bands and quantized anomalous Hall effect at odd fillings in hexagonal boron nitride (h-BN) aligned twisted bilayer graphene (TBG)[19,20] and ABC trilayer graphene[9] as well as twisted monolayer-bilayer graphene[11,12], while spin polarization is critical to the formation of correlated insulator at $v = 2$ in TDBG[5–8,21,22]. Intuitively, one might ask if it is possible to drive the polarized states from one flavor to the other, and vice versa. This interesting question has been partially targeted in recent observations of the Pomeranchuk effect in TBG which involves a transition of isospin from unpolarized to polarized states[23,24]. However, it remains open whether one could achieve a tunable transition of polarized states from one isospin to the other, and moreover whether this transition will lead to new phase of matter.

Among many twisted graphene-based moiré systems, TDBG is a promising target to address this question based on the following concerns. First, TDBG is a strongly displacement field ($D$) tunable moiré system[25,26]. The displacement field can in-situ tune the relative strength of Coulomb interactions, relative to its kinetic energy, thus acts an extra knob beyond the twist angle to tune the band structure and topological properties of TDBG. Second, correlated insulator at half filling in TDBG are spin polarized[5–7], and it emerges over a wide range of twist angle from ~1.1° to ~1.5°, with a finite $D$ varying from ~0.2 to ~0.5 V/nm. The symmetry breaking instability occurs at the boundary between spin-polarized and unpolarized states in TDBG, and it give rise to quantum critical behaviors[22]. Besides, TDBG might, in principle, host valley-polarized states as the ground states since the perpendicular magnetic field could induce a competition between spin and valley by spin and orbital Zeeman effects[27], thus resulting to an isospin transition from one to the other.

In this work, we report a study of isospin polarizations in TDBG. We fabricate ultra-clean TDBG devices with gold top gate and graphite bottom gate. We push the displacement field to its limit, and importantly we unveil a transition from spin polarization to valley polarization when $D$ approaches a critical field $D^*$. The transition is accompanied by new phases of matter, i.e. a valley polarized Chern insulator state at $v = 2$ in the electron side and a valley polarized Fermi surface around $v = -2$ in the hole side, which never realized in previous twisted graphene-based moiré system. The valley Chern insulator shows a well quantized Hall conductance plateau at $2e^2/h$ and correspondingly a vanishing longitudinal component. The valley polarized Fermi surface shows a series of quantized Landau levels (LLs) with $v_{LL} = 0$, ±1, ±2, ±3, ±4 and others in the landau fan diagram.

## Results and discussion
### General information of TDBG devices.
The TDBG devices are fabricated by the cut and stack technique[28]. A big flake of AB-stacked bilayer graphene is firstly cut into two pieces then stacked up with a twist angle of 60° + $\theta$, so called AB-BA TDBG, as shown in Fig. 1a, where $\theta$ is ~1.3° to ensure a strong electron correlation of TDBG[5]. Compared to the previously studied TDBG

with a twist angle of $\theta$ (AB-AB TDBG), AB-BA TDBG has similar band structure yet different topological properties according to the theoretical calculations[26,29–32]. The stacked samples are ultra-clean with a bubble-free area over a length scale of ~20 μm, as shown in Fig. 1a and Supplementary Fig. 5a. These devices have a dual-gate configuration with a graphite bottom gate and gold top gate, which allows independent tuning of the carrier density $n$ and $D$. Here, $n = (C_{BG}V_{BG} + C_{TG}V_{TG})/e$ and $D = (C_{BG}V_{BG} - C_{TG}V_{TG})/2\varepsilon_0$, where $C_{BG}$ ($C_{TG}$) is the geometrical capacitance per area for bottom (top) gate, $e$ is the electron charge, and $\varepsilon_0$ is the vacuum permittivity. These devices show a good quality with the angle inhomogeneity <0.01° within 2 μm size and mobility on order of $10^5$ cm²/(V·s) in Fig. 1b, c and Supplementary Fig. 5.

We perform cryogenic magneto-transport measurements at a base temperature of $T = 1.8$ K. Figure 1d shows the longitudinal resistance ($R_{xx}$) as a function of filling factor ($v$) and $D$ at $B = 0$ T. The filling factor is defined as $v = 4n/n_s$, corresponding to the number of carriers per moiré unit cell. Here, $n_s = 4/A \approx 8\theta/(\sqrt{3}a^2)$, where $A$ is the area of a moiré unit cell, $\theta$ is twisted angle, and $a$ is the lattice constant of graphene. In Fig. 1d, two evident resistance peaks at $v = -4$ and 4 correspond to the moiré band gap, and the resistance peak at $v = 0$ indicates a band gap opening between the conduction band (CB) and the valance band (VB) which results from $D$ induced inversion symmetry breaking[33]. Correlated insulators at half filling $v = 2$ are observed at a medium $D$ from 0.3 to 0.6 V/nm in device D1 (Fig. 1d) and a similar $D$ from 0.2 to 0.5 V/nm in device D2 (Fig. 2a). These correlated insulators are spin polarized as evidenced from its positive in-plane magnetic field response (Fig. 2b, d) and a corresponding Zeeman spin $g$ factor $g_s$ of ~2.25 (Fig. 2f). Similarly, a spin-polarized insulator is developed at $v = 1$ in Fig. 2b and d, with $g_s$ of ~2.35 (Fig. 2f). All these features both at zero magnetic field and in-plane magnetic field are in agreements with previous results of AB-AB TDBG[5–7], which are also in line with the identical band structure between AB-BA and AB-AB TDBG[26].

**Valley polarized correlated insulators at high displacement fields.** We observe new correlated insulators in the hole side at $v = -2$ when $D$ is sufficiently large. The new correlated insulators are manifested as new resistance peaks in the hole side at $v = -2$, as shown in Fig. 1e when $|D| > 0.75$ V/nm for device D1 and in Fig. 2c when $|D| > 0.6$ V/nm for device D2, in the presence of a finite perpendicular magnetic field. It is unexpected as the valance band becomes more and more dispersive and entangles with remote bands with the increase of $|D|$, and it demands an important role played by electron correlation. In principle, valley degeneracy can be lifted in a perpendicular magnetic field[34,35], as shown in Fig. 1f. Considering a Bloch electron is subjected to a weak magnetic field, the energy can be expressed as $\varepsilon_{N,\sigma,\tau}(\mathbf{k}, B) = \varepsilon_{N,\tau}(\mathbf{k}) + \sigma Sg_s\mu_B B + m_{N,\tau}(\mathbf{k})B$, where the second (third) term is the spin (orbital) Zeeman energy. Here, $N$ is the band index, $\sigma = \pm 1$ is the spin index, $S = 1/2$ is spin quantum number, and $\tau = \pm 1$ is the valley index. In the third term, $m_{n,\tau}(\mathbf{k})$ corresponds to the orbital magnetization, and it is opposite in different valley K and K' according to the time reversal symmetry. Therefore, the valley degeneracy can be lifted with the orbital Zeeman effect. The last two terms can be expressed as $g\mu_B B$, where $g$ is the effective $g$ factor ($g = S^*g_s = 1$ for pure spin Zeeman effect).

We perform Zeeman effect measurements for the new correlated insulator at $v = -2$. The resulting Zeeman thermal gap as a function of $B_\perp$ is plotted in Fig. 1g, and from which we obtain effective Zeeman $g \approx 13.6$. This insulator is also not related to spin degree of freedom, as it only response to perpendicular magnetic field and it is featureless even at $B_\| = 9$ T (Fig. 2b).

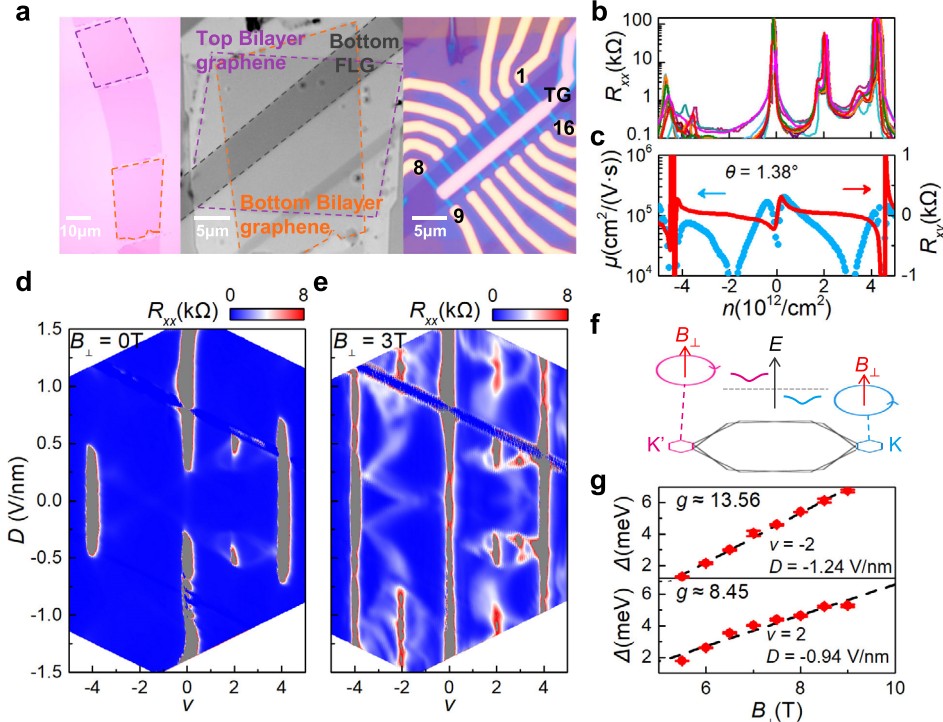

**Fig. 1 Valley polarized correlated insulating states in AB-BA TDBG. a** Optical microscope images of the fabrication process of device D1($\theta = 1.38°$). The bilayer graphene, ABBA-TDBG and the dual-gate device are presented in turn from left to right. The bottom gate is few layers graphite (FLG) and the top gate (TG) is Ti/Au. The scale bar is shown in figure. **b** Four-terminal longitudinal resistance versus carrier density $n$ at $D = -0.46$ V/nm between every two adjacent bars from 1 to 16. **c** Hall mobility and Hall resistance versus carrier density at $D = 0$ V/nm. **d, e** Longitudinal resistance $R_{xx}$ as a function of filling factor $v$ and displacement field $D$ in device D1. Left and right figures correspond to the transport data measured at $B_\perp = 0$ T and $B_\perp = 3$ T, respectively. **f** Schematic of the valley polarization. The blue and pink circles represent the orbital magnetization of K and K' valley, respectively. Different direction of arrows indicates the orbital magnetizations of two valleys are opposite. The blue and pink curves correspond to the valley polarized energy band induced by the orbital Zeeman effect under the perpendicular magnetic field. **g** Thermal activation gaps versus perpendicular magnetic fields. The top figure shows the energy gap at $v = -2$ and $D = -1.24$ V/nm, and the bottom figure shows the energy gap at $v = 2$ and $D = -0.94$ V/nm. Error bars are estimated according to the uncertainty at the thermal activation region.

Thus, we assign the correlated insulator at $v = -2$ as a valley-polarized insulator.

Moreover, we also observe a new correlated resistance peak at electron side $v = 2$ when $|D| > 0.9$ V/nm, aside from the conventional spin correlated insulator at $v = 2$ when $D$ is at mediate range from 0.3 to 0.6 V/nm, as shown in Fig. 1e. At a higher $B_\perp$, the insulator at the higher $D$ is much more pronounced (Fig. 1e), while the spin-polarized insulator at the lower $D$ becomes weaker and weaker and eventually disappear at $B_\perp > 5$ T (Supplementary Fig. 13). We plot the gap for the new insulator at higher $D$ as a function of $B_\perp$ in Fig.1g, and once again we get $g \approx 8.5$, indicating a dominating role played by valley instead of spin. Thus, the correlated insulator at $v = 2$ when $|D| > 0.9$ V/nm is also valley polarized, similar to that at $v = -2$.

**Competition of isospin polarization between spin and valley.** The disappear of spin-polarized insulator and simultaneously the rise of the valley-polarized insulator at $v = 2$ with increase $B_\perp$ suggest a competition between spin and valley polarizations[35,36]. Such a competing scenario is even more pronounced in the valence band at $v = -2$. The evolution of $R_{xx}$ at $v = -2$ with $B_\parallel$ is studied at a fixed $B_\perp = 6$ T in D2, as shown in Fig. 2e. Note that the orbital Zeeman splitting energy remains almost unchanged and only the spin Zeeman effect need to be considered in this situation. The peak resistance at $v = -2$ decreases as $B_\parallel$ increases, and it shows an unconventional insulating behavior where $R(T)$ doesn't present a well thermal activation behavior. Instead, it

could be divided into two parts, i.e. a strong $B_\parallel$ dependent and $T$ sensitive insulating behavior at $T < 10$ K and an almost $B_\parallel$ independent insulating state at $T > 16$ K. According to the thermal activation behavior $R_{xx} = R_0\exp(\Delta/(2k_BT))$, where $k_B$ is the Boltzmann constant, we estimated the thermal energy gaps ($\Delta$) from $R(T)$ at $T < 10$ K in Fig. 2g. We can see that $\Delta$ decreases as $B_\parallel$ increases, and the spin $g$ factor $g_s \approx -1.26$. The negative sign of $g_s$ indicates that spin Zeeman effect tend to close the gap of the valley polarized insulator at $v = -2$, a strong evidence of competing instead of cooperating between spin and valley polarization. Similar competing behaviors and negative $g_s$ are also observed at $v = 2$ in device D1(Supplementary Fig. 7).

**Valley Chern insulator with $C = 2$ emanating from $v = 2$.** The topological nature of the valley polarized moiré flat band can be revealed in the Landau fan diagram[18,37–42]. Figure 3a shows the longitudinal resistance ($R_{xx}$) as a function of $v$ and $B_\perp$ at $D = 0.8$ V/nm and $T = 1.8$ K. An obvious wedge-shaped $R_{xx}$ minimum emanating from $v = 2$ develops along the line with a slope of $dn/dB = Ce/h$, where $C$ is equal to 2. The vanishing $R_{xx}$ comes together with a plateau of Hall resistance ($R_{xy}$), with an onset magnetic field of $B_\perp = 4.2$ T. The perfect quantization of $R_{xy}$ as well as corresponding zero $R_{xx}$, i.e. $\sigma_{xy} = 2e^2/h$ with $\sigma_{xx} \sim 0$, is demonstrated in Fig. 3b at a fixed $B_\perp = 7.3$ T and Figure 3c at a fixed carrier density of $v = 2.46$. Here, the conductance $\sigma_{xx}$ and $\sigma_{xy}$ are given by: $\sigma_{xx} = a_r * R_{xx}/((a_r * R_{xx})^2 + R_{xy}^2)$,

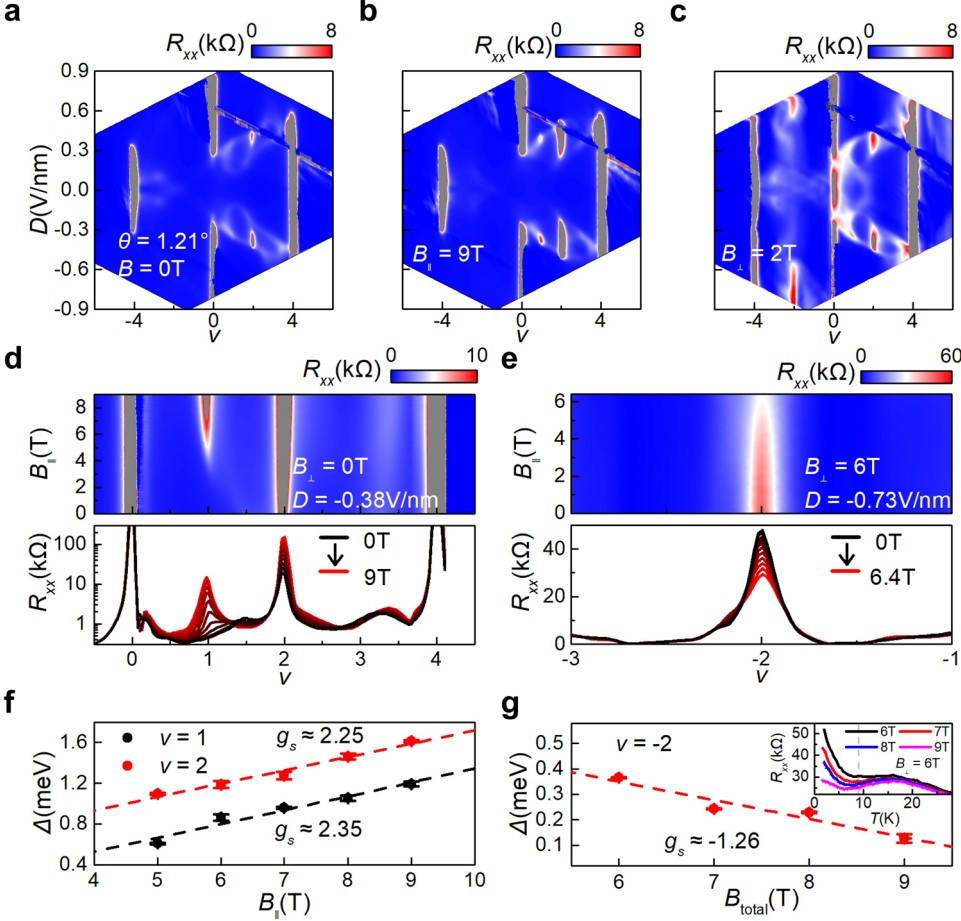

**Fig. 2 Competition between spin and valley polarization. a–c** Longitudinal resistance $R_{xx}$ as a function of filling factor $v$ and displacement field $D$ of device D2 ($\theta = 1.21°$) at $B = 0$ T, $B_{\parallel} = 9$ T and $B_{\perp} = 2$ T, respectively. **d** Top, longitudinal resistance $R_{xx}$ as a function of filling factor $v$ and in-plane magnetic field $B_{\parallel}$ at $D = -0.38$ V/nm. Bottom, line cuts of $R_{xx}$ ($v$, $B_{\parallel}$) from $B_{\parallel} = 0$ to $B_{\parallel} = 9$ T. **e** Top, longitudinal resistance $R_{xx}$ as a function of filling factor $v$ and in-plane magnetic field $B_{\parallel}$ at $D = 0.73$ V/nm and $B_{\perp} = 6$ T. Bottom, line cuts of $R_{xx}(v, B_{\parallel})$ from $B_{\parallel} = 0$ to $B_{\parallel} = 6.4$ T. **f** Thermal activation gaps versus $B_{\parallel}$ at $v = 1$ and $v = 2$ corresponding to the insulating states in **d**. The spin g factor can be extracted from the linear fitting with the spin Zeeman effect, $\Delta \sim 2 \times Sg_s\mu_BB$ and $S = 1/2$. **g** Thermal activation gaps versus total magnetic field at $v = -2$ corresponding to the insulating state in **e**. All Error bars are estimated according to the uncertainty at the thermal activation region. Inset, temperature dependence of $R_{xx}$ under the tilted magnetic field. The perpendicular magnetic field is fixed at 6 T and $B_{total}$ increases from 6 T to 9 T.

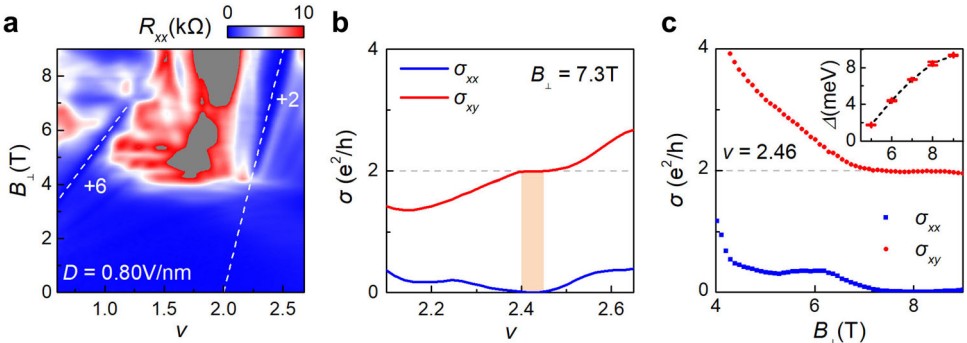

**Fig. 3 Valley polarized Chern insulator in CB. a** Longitudinal resistance $R_{xx}$ as a function of filling factor $v$ and perpendicular magnetic field $B_{\perp}$ at $D = 0.8$ V/nm in device D2. White dash lines correspond to the LL with $v_{LL} = +6$ emanating from $v = 0$ and the $C = 2$ Chern insulator emanating from $v = 2$, respectively. **b** Line cuts of $\sigma$ ($v$, $B_{\perp}$) at $B_{\perp} = 7.3$ T. The plateau within the orange color area indicates a well-quantized Chern insulator with $\sigma_{xx} = 0$ and $\sigma_{xy} = 2e^2/h$. **c** Line cuts of $\sigma$ ($v$, $B_{\perp}$) at $v = 2.46$. Inset, thermal activation gaps of the Chern insulator versus perpendicular magnetic field. Error bars are estimated according to the uncertainty at the thermal activation region.

$\sigma_{xy} = R_{xy}/((a_r * R_{xx})^2 + R_{xy}^2)$, where $a_r$ is the aspect ratio of the device. Similar quantization with $C = 2$ is also observed in other three devices with both AB-BA and AB-AB stacking, as shown in the Supplementary Fig. 9–11. The observed quantization with $C = 2$ is in good agreement with our theoretic calculation of the valley polarized moire Chern band in TDBG (see Supplementary Note 1 for details), where $C_v = 1$ and the total Chern number $C = 2*C_v$ by including two-fold spin degeneracy.

The magnetic field dependent $\triangle$ for the $C = 2$ valley Chern insulator is illustrated in the inset of Fig. 3c. The gap $\triangle$ increases with $B_\perp$ and tends to saturate at high $B_\perp$, being consistent with the mechanism that the valley polarization becomes stronger with the increased $B_\perp$. We compare the gap of valley Chern insulator with that of the first LL ($\nu_{LL} = +6$) emanating from $\nu = 0$ at $B_\perp = 6$ T (Supplementary Fig. 8), and it turns out that the energy gap of the former (~4.3 meV) is much larger than that of the latter (~1.6 meV). The valley Chern insulator is also clearly distinguished from the faint features of LLs (with $\nu_{LL} = +1, +3, +4$) emanating from $\nu = 2$ in the fan diagram of Fig. 3a, where the $R_{xx}$ dip with $C = 2$ reaches zero while the rest not. The distinguishment is further illustrated in the perfect quantization of Hall conductance with $C = 2$ in Fig. 3b, c.

**Fermi surface reconstruction and Landau quantization around $\nu = -2$.** In contrast to the topological valley subband in CB, it is a trivial valley polarized subband with $C = 0$ in VB according to our calculations with Hartree-Fock approximation at a large $D$, and the VB is generally more dispersive than CB as $D$ is increased (see Supplementary Note 1 for details). It goes through a series of transformations due to enhanced valley polarization as $B_\perp$ is increased. Figure 4 illustrates typical Fermi surface reconstructions driven by valley polarization at different magnetic field when a large $D = -0.73$ V/nm is applied. Essentially, the reconstruction is captured in the change of Hall filling factors $\nu_H = 4n_H/n$ to moiré band filling factor $\nu$, where Hall carrier density $n_H = B_\perp/eR_{xy}$.

At $B_\perp = 0.8$ T, the magnetic field is too small to reconstruct the energy band, and the $\nu_H$ in Fig. 4c varies linearly, indicated by two blue dashed lines of $\nu_H = \nu$ and $\nu_H = \nu + 4$ near the charge neutrality point (CNP) and moiré band edge, respectively. van Hove singularity (VHS) is indicated by the diverging Hall carrier density near $\nu = -2$, which suggests a Lifshitz transition of the Fermi surface[39,43]. Note that VHS locates at around $\nu = -2$, corresponding to the flat band of the highest VB, based on our band structure calculations from the continuum model (Supplementary information). The position of VHS changes with the increased electric field, and it follows the white contour line in Fig. 4b. At $B_\perp = 4$ T, the Fermi surface of VB is greatly reconstructed (Fig. 4e–g). The divergent carrier densities are reset to zero at $\nu = -2$ and extend to both sides along the line of $\nu_H = \nu + 2$. In this case, a correlation driven energy gap replaces the VHS and separates the VB into two valley-polarized subbands with new VHSs near $\nu = -1$ and $\nu = -3$.

The reconstruction of VB is reminiscent of the stoner criterion in ferromagnetic metal[44]. The paramagnetic phase becomes of instability if $U*g(E_F) > 1$, where $U$ and $g(E_F)$ are the Coulomb repulsion and density of states (DOS) at the Fermi surface, respectively. Hence, similar to the stoner mechanism, the occupancy of VB is redistributed between two valley flavors facilitated by the divergent DOS at VHS, and then the orbital Zeeman effect separate the two valley-polarized subbands with the increased $B_\perp$.

The complete Landau fan diagram is present in Fig. 4i. LLs emanating from $\nu = -2$ are observed with $\nu_{LL} = \pm1, \pm2, \pm3, \pm4, -5,$

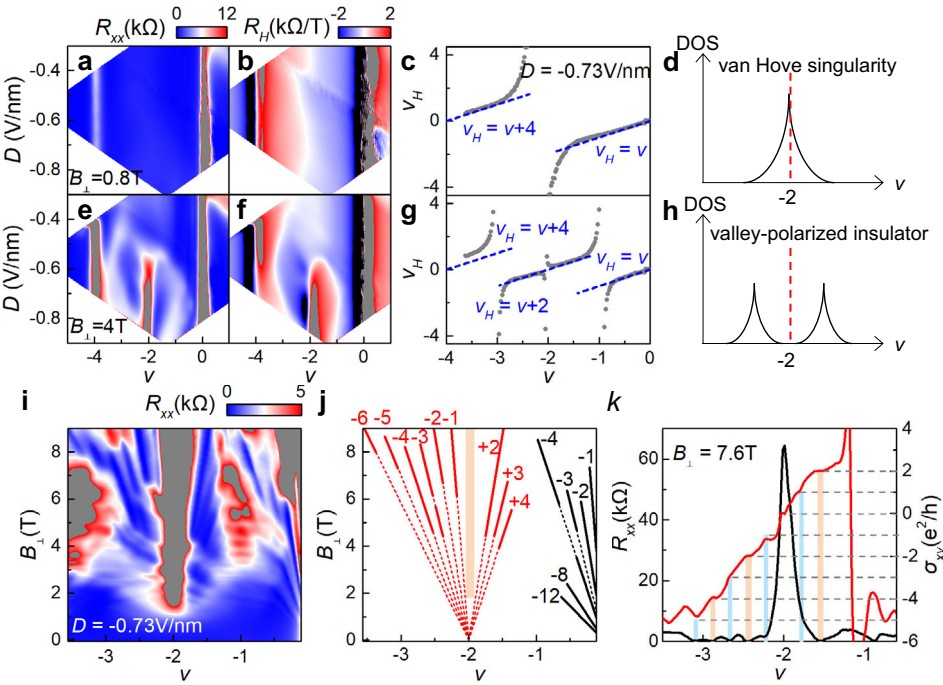

**Fig. 4 Fermi surface reconstruction and Landau fan diagram of VB. a, b, e, f** Longitudinal resistance $R_{xx}$ and Hall coefficient $R_H$ as a function of filling factor $\nu$ and displacement field $D$ at $B_\perp = 0.8$ T and 4 T, respectively (device D2). **c, g** Line cuts of mapping at $D = -0.73$ V/nm show Hall filling factor $\nu_H$ as a function of filling factor $\nu$. **d, h** Schematics of density of states at $B_\perp = 0.8$ T and 4 T, respectively. The red dashed line corresponds to $\nu = -2$. **i** Longitudinal resistance $R_{xx}$ as a function of filling factor $\nu$ and perpendicular magnetic field $B_\perp$ at $D = -0.73$ V/nm. **j** Schematic of LLs shown in **i**. Red lines correspond to the LLs emanating from $\nu = -2$, and black lines correspond to the LLs emanating from $\nu = 0$. **k** Line cuts show the well-quantized $\sigma_{xy} = \pm2e^2/h, -4e^2/h$ with almost zero $R_{xx}$ (orange color bars) and incipiently quantized $\sigma_{xy} = \pm e^2/h, -3e^2/h, -5e^2/h$ with finite $R_{xx}$ at $B_\perp = 7.6$ T (blue color bars).

−6, indicating the spin and valley degeneracy are completely lifted. The even LLs are derived from the valley-polarized subbands, while the odd LLs correspond to the quantum hall ferromagnets[45,46] which polarize the spin flavors. The difference is reflected in the energy scale between them. As shown in Fig. 4k, even LLs with $\nu_{LL} = \pm 2, -4$ reveal well quantized Hall plateaus and zero $R_{xx}$. However, odd LLs with $\nu_{LL} = \pm 1, -3, -5$ show finite $R_{xx}$ and incipient Hall plateau which indicates incomplete quantization. The perpendicular magnetic field plays a key role on the reconstruction of VB. It not only interacts with opposite orbital magnetic moments at different valleys, promoting the formation of valley-polarized states, but also makes the carriers do the cyclotron motion in valley-polarized subbands.

We have observed new correlated insulators evolved from spin-polarized states to valley-polarized states in TDBG. The transition of the isospin polarization is a result of the competition between spin and valley, driven by the displacement field and magnetic field. Moreover, we have unveiled the unique topology of the TDBG, including a quantized valley Chern insulator with $C = 2$ emanating at $\nu = 2$ in the electron side and a valley polarized yet topologically trivial Fermi surface with $C = 0$ around $\nu = -2$ in the hole side. It is worth mentioning that our devices are of high quality and state-of-the-art clean, with twist angle inhomogeneity <0.01°. Our studies shed light on the importance of the displacement field to tune the band topology, and our results could enrich the current understanding of the TDBG system and provide references for other twisted or non-twisted multilayer systems as well.

## Methods

**Estimation of the twisted angle.** The top and bottom gate capacitance can be extracted from the Landau fan diagram and dual-gate voltage mapping. We calculate the carrier density $n$ and the electric displacement field $D$ according to the capacitances and transform the $R_{xx}$ ($V_b$, $V_t$) mapping to the $R_{xx}$ ($n$, $D$) mapping. According to the formula, $n_s = 4/A \approx 8\theta/(\sqrt{3}a^2)$, the twisted angle $\theta$ can be extracted from the location of superlattice resistance peak in the $R_{xx}$ ($n$, $D$) mapping. The obtained twist angle is further corrected by the Brown-Zak oscillations in the Landau fan diagram. As shown in Supplementary Fig. 10, $R_{xx}$ shows dips at $\Phi/\Phi_0 = 1/q$, where $\Phi = AB_\perp$ is the magnetic flux per moiré unit cell, $\Phi_0 = h/e$ is the magnetic flux quantum, and $q$ is a positive integer.

**Estimation of the Zeeman g factor.** The Zeeman splitting energy, including the contributions from both spin and orbital magnetization, can be expressed as $\triangle \sim 2g\mu_B B$, where $g$ corresponds the effective $g$ factor. The value of $g$ can be extracted from the linear fitting of thermal activation energy gap versus magnetic field. Under the in-plane magnetic field, only the spin Zeeman effect needs to be considered, hence $\triangle \sim 2g\mu_B B = 2Sg_s\mu_B B$, where $S = 1/2$ is the spin quantum number and $g_s$ is spin $g$ factor. To clearly show the role of spin, we present our results using $g_s$ instead of $g$ in this situation.

**Calibration of the magnetic field direction.** The change of the magnetic field direction is achieved by rotating the sample with the attocube stage. Then the direction is further calibrated by measuring Hall resistance at different rotating angle. We measured the Hall resistance at $V_{tg} = V_{bg} = 0$ V from −3 T to 3 T in perpendicular and parallel direction, as shown in Supplementary Fig. 15. The response of Hall resistance totally come from the cyclotron motion of the electron under the perpendicular magnetic field. Hence, the value of the Hall resistance under the parallel magnetic field can be used to calibrate the direction of the magnetic field. The $R_{xy}$ is about −30 ohm at $B_\parallel = 3$ T, which is equivalent to the $R_{xy}$ at $B_\perp = 0.03$ T. To better visualize the difference, we shrink X-axis of the parallel magnetic field by ~100 times, i.e. −30 mT to 30 mT, while keeping the X-axis of the perpendicular direction unchanged; the almost coincident lines suggest the residual perpendicular component is ~30 mT at $B_\parallel = 3$ T, and thus the error of parallel magnetic field is about 1%. Since it needs a large perpendicular magnetic field to realize the valley-polarized states, thus the small error has no effect on our experiment results.

## Data availability

The data that support the findings of this study are available from the corresponding authors upon reasonable request.

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

## Acknowledgements

We acknowledge supports from the National Key Research and Development Program (Grant No. 2020YFA0309600), National Natural Science Foundation of China (NSFC, Grant Nos. 61888102, 11834017, 12074413), the Strategic Priority Research Program of CAS (Grant Nos. XDB30000000& XDB33000000) and the Key-Area Research and Development Program of Guangdong Province (Grant No. 2020B0101340001). Y. H acknowledge support from the National Key Research and Development Program of China (Grant No. 2019YFA0308000) and the National Natural Science Foundation of China (NSFC, Grants No. 62022089). K.W. and T.T. acknowledge support from the Elemental Strategy Initiative conducted by the MEXT, Japan (Grant No. JPMXP0112101001), JSPS KAKENHI (Grant Nos. 19H05790, 20H00354 and 21H05233) and A3 Foresight by JSPS.

## Author contributions

W.Y. and G.Z. supervised the project; L.L., W.Y., G.Z. designed the experiments; L.L., Y.C. fabricated the devices with assistance from Y.Y., J.T., Y.J., J.T., R.Y., D.S.; L.L., Y.C., C.S., Y.H. performed the magneto-transport measurement; S.Z. and J.L. performed the calculations; K.W. and T.T. provided hexagonal boron nitride crystals; L.L., W.Y., G.Z., analyzed the data; W.Y., L.L., G.Z. wrote the paper with the input from all the authors.

## Competing interests

The authors declare no competing interests.
