## [Peer Review File · Nature Communications]

Isospin competitions and valley polarized correlated insulators in twisted double bilayer grapheneREVIEWER COMMENTS

Reviewer #1 (Remarks to the Author):

In this manuscript, the authors fabricated ABBA-stacked twisted double bilayer graphene devices, and found spin or valley polarized insulators and their interplay via electrical transport measurements by controlling electric displacement fields and external magnetic fields. Both topologically nontrivial valley Chern insulator states and trivial valley polarized insulators are found, depending on the charge density doping. The devices are of high quality, the presented data are clean, and the manuscript is generally well-organized. There are a few points that the authors may want to clarify to strengthen their arguments. Therefore, I would recommend its publication in Nature Communications, if the authors can provide adequate answers to the comments below.

1. The authors claim that the observed valley polarized correlated insulators appear only at large displacement field and under perpendicular magnetic field, while not present under a large magnetic field in the in-plane direction. While the latter part of the reasoning (large parallel magnetic field) is shown for device D2 in Fig.2 so that it is shown for the hole-doped valley polarized insulator, such ν -D map of device D1 at large parallel field (at zero perpendicular field) is missing. The authors should provide the full map for D1 either in the main figure, as the reasoning for the valley polarization depends heavily on the absence of the state under large parallel field.
2. Device D1 shows the new correlated insulator states at large D appearing in both hole and electron doped sides at $B=3T$ in the perpendicular direction. These states appear at approximately the same D for this device. However, device D2 only shows the correlated insulator at large D only on the hole-doped side. I understand that the gate ranges are more limited in this device than in D1, but given that they appear around the same D, they should also be observed in the given D range in the map. The authors should comment on this difference.
3. The whole interpretation on whether something is spin or valley polarized, as well as their claimed "competitions," entirely depends on the appearance and disappearance of these states under perpendicular or parallel magnetic fields and the extracted gap analysis. Therefore, accurate calibration of the field direction is necessary. Even a small tilt between the field direction and the same direction can cause unwanted tilted fields. The authors should provide the methods and data on how they calibrated the tilt in the magnetic fields, and the precision of their calibration in the scale of mT.
4. Even the accurately calibrated parallel magnetic fields can induce orbital effects due to the finite thickness of the four-layered sample. Can the authors comment on this scale and how they eliminate such effect to analyze purely spin Zeeman factors?

Reviewer #2 (Remarks to the Author):

The manuscript by L. Liu et al. reports the observation of new correlated insulator states under relatively large displacement fields (D) in AB-BA stacked twisted double bilayer graphene (TDBG). The authors attribute the observations to the competition of isospin polarization between spin and valley. They also show the measured Landau Fan diagram around the new correlated insulator states. Overall, I think the manuscript would be of great interest to researchers in the fields of graphene-based moiré superlattices and also other moiré systems. It demonstrates a feasible way to realize isospin control in TDBG and could enrich the current understanding of the TDBG system. I would be happy to recommend the paper's publication in Nature Communications if the authors could address the following comments.

1. Is the valley polarized correlated insulator states under relatively large D unique for the AB-BA stacked TBDG? Results from an AB-AB stacked device measured under similar D range (about -1.25 V/nm to 1.25 V/nm at $\nu=\pm 2$) would be very helpful.
2. In Fig. 2(d), 2(e), and 3(a), it seems there are abrupt changes of measured resistance (oblique lines on these contour plots) in the electron side under positive D . Are they due to bad contacts?
3. For the g factors obtained in Fig. 1(g) and 2(g), are they symmetric with D ? It is better to show results at both positive D and negative D .
4. The authors attribute the disappear of spin polarized insulator at $\nu=2$ with increasing B_{\perp} to the competition of isospin polarization between spin and valley polarizations. Actually, this observation may also relate to the B_{\perp} induced band reconstruction. To clarify it, the authors should show similar analysis as Fig. 2(g) also for the valley polarized insulator at $\nu=2$.
5. The temperature dependence shown in the inset of Fig. 2(g) need to be further discussed. Is this unusual temperature dependence related to the spontaneously symmetry broken? BTW, in line 130, the ' $R(T)$ ' should be ' R_{xx} '.
6. The authors claim their TBDG devices are ultra-clean. However, from the measured Landau fan diagrams, the data quality seems not very high. For example, the onset of quantum oscillations is about 2-4 T, which is lower than previous studies, e.g. $\sim 0.5-1$ T in X. Liu et al. Nature 583 221 (2020). Also, the insulating state shown in Fig 3(a) between $\nu=1.0$ to 2.0 looks very broad, which is likely due to some kinds of disorder effects.
7. A minor comment. The terminologies used in the introduction part need to be further polished. For example, it is more accurate to use 'graphene-based moiré system' instead of 'twisted moiré system' in the 1st to 3rd paragraph, since nowadays there are other moiré systems beyond graphene, e.g. TMDc moiré systems. And some sentences in the manuscript are only correct for the 'graphene-based moiré system', such as line 40-41 on page 2, "Each conduction and valence moiré band has four isospin flavors, involving spin and valley degree of freedoms".

The main changes in the revised manuscript

1. We have added the results of the AB-AB stacked device and competitions between spin and valley at $\nu = 2$ in device D1 in revised manuscript.
2. We have included 1 additional reference (Ref. 36) and updated 1 reference (Ref. 35) in the revised manuscript.
3. We have added the calibration method of the magnetic field direction and moved the method part from supplementary information to revised manuscript.
4. We have included the Fig. R3, Fig. R4, Fig. R6 and Fig. R8 in the supplementary information.

Reviewer #1 (Remarks to the Author):

In this manuscript, the authors fabricated ABBA-stacked twisted double bilayer graphene devices, and found spin or valley polarized insulators and their interplay via electrical transport measurements by controlling electric displacement fields and external magnetic fields. Both topologically nontrivial valley Chern insulator states and trivial valley polarized insulators are found, depending on the charge density doping. The devices are of high quality, the presented data are clean, and the manuscript is generally well-organized. There are a few points that the authors may want to clarify to strengthen their arguments. Therefore, I would recommend its publication in Nature Communications, if the authors can provide adequate answers to the comments below.

Response: We thank the reviewer for the very careful reading of our manuscript and constructive suggestions. We are delighted to find that the reviewer found our devices are of “**high quality**”, the data “**clean**”, and the manuscript “**well-organized**”. We appreciate the reviewer for recommending the publication in Nature Communications.

1. The authors claim that the observed valley polarized correlated insulators appear only at large displacement field and under perpendicular magnetic field, while not present under a large magnetic field in the in-plane direction. While the latter part of the reasoning (large parallel magnetic field) is shown for device D2 in Fig.2 so that it is shown for the hole-doped valley polarized insulator, such ν - D map of device D1 at large parallel field (at zero perpendicular field) is missing. The authors should provide the full map for D1 either in the main figure, as the reasoning for the valley polarization depends heavily on the absence of the state under large parallel field.

Response: We thank the reviewer for the critical comment and constructive suggestions.

We have re-cooled down the device D1 and carried out a measurement of longitudinal resistance R_{xx} as a function of filling factor ν and displacement field D at $B_{\parallel} = 9\text{T}$, as shown in Fig. R1. The valley polarized insulators don't appear in this situation, which further supports our conclusion about valley polarization.

Fig. R1. A color mapping of $R_{xx}(v, D)$ at $B_{\parallel} = 9T$ in device D1.

2. Device D1 shows the new correlated insulator states at large D appearing in both hole and electron doped sides at $B=3T$ in the perpendicular direction. These states appear at approximately the same D for this device. However, device D2 only shows the correlated insulator at large D only on the hole-doped side. I understand that the gate ranges are more limited in this device than in D1, but given that they appear around the same D , they should also be observed in the given D range in the map. The authors should comment on this difference.

Response: We thank the reviewer for the comment. We agree with the reviewer that the valley polarized correlated states at $\nu = 2$ and that at $\nu = -2$ should in principle appear at around the same D . However, variations between samples are expected, which might give different orbital Zeeman g factors and thus require different onset magnetic fields for the emergence of valley polarized correlated states.

We have carried out more measurements for device D2, as shown in Fig. R2. While the $\nu = -2$ valley polarized insulator appears at a small magnetic field $B_{\perp} = 2T$, the $\nu = 2$ valley polarized insulator shows up at a larger magnetic field $B_{\perp} = 6T$. Besides, the two insulators at $\nu = -2$ and $\nu = 2$ do indeed appear at the similar range of displacement field. These additional data from device D2 agrees with those in device D1.

Fig. R2. Valley polarized insulators in CB of device D2. (a), (b) Longitudinal resistance R_{xx} as a

function of filling factor ν and displacement field D at $B_{\perp} = 2T$ and $B_{\perp} = 6T$, respectively.

3. The whole interpretation on whether something is spin or valley polarized, as well as their claimed “competitions,” entirely depends on the appearance and disappearance of these states under perpendicular or parallel magnetic fields and the extracted gap analysis. Therefore, accurate calibration of the field direction is necessary. Even a small tilt between the field direction and the same direction can cause unwanted tilted fields. The authors should provide the methods and data on how they calibrated the tilt in the magnetic fields, and the precision of their calibration in the scale of mT.

Response: We thank the reviewer for the comment. In the revised manuscript, we have included the method for calibrating magnetic field, which is done by comparing the Hall resistance in the two directions. As shown in Fig. R3, we measured the Hall resistance at $V_{tg} = V_{bg} = 0V$ from $-3T$ to $3T$ in perpendicular and parallel direction, respectively. The response of Hall resistance totally come from the cyclotron motion of the electron under the perpendicular magnetic field. Hence, the value of the Hall resistance under the parallel magnetic field can be used to calibrate the direction of the magnetic field. The R_{xy} is about -30 ohm at $B_{\parallel} = 3T$ (Fig. R3(b)), which is equivalent to the R_{xy} at $B_{\perp} = 0.03T$. To better visualize the difference, we shrink X axis of the parallel magnetic field by ~ 100 times, i.e. $-30mT$ to $30mT$ in Fig.R3(c), while keeping the X axis of the perpendicular direction unchanged. As shown in Fig. R3(c), the almost coincident lines suggest the residual perpendicular component is $\sim 30mT$ at $B_{\parallel} = 3T$, and thus the error of parallel magnetic field is about 1%. Since it needs a large perpendicular magnetic field to realize the valley polarized states, thus the small error has no effect on our experiment results.

In the revised manuscript, we have added the Fig. R3 as Fig. S15.

Fig. R3. Calibration of the field direction. (a)-(c) Hall resistance R_{xy} versus magnetic field B in perpendicular and parallel directions. In figure (c), the parallel magnetic field are reduced by ~ 100 times to compare with the perpendicular magnetic field.

4. Even the accurately calibrated parallel magnetic fields can induce orbital effects due to the finite thickness of the four-layered sample. Can the authors comment on this scale and how they eliminate such effect to analyze purely spin Zeeman factors?

Response: We thank the reviewer for the comment. Yes, the in-plane orbital effect is inevitable. The

paper by Lee, J. Y. et al. [Nature Communications 10, 5333 (2019)] discussed the in-plane orbital effect. They used the perturbation theory to consider the contribution of this part:

$$\varepsilon_{N,\tau}(\mathbf{k}, B) = \varepsilon_{N,\tau}(\mathbf{k}) + g_{N,\tau}^{xy}(\mathbf{k})\mu_B B$$

where $g_{N,\tau}^{xy}(\mathbf{k})$ is the in-plane orbital g factor. They calculated the in-plane and out-plane orbital g factor, and the former is much smaller than the latter. Actually, they discussed the influence of in-plane orbital effect on the spin-polarized insulator and gave a 20~50% deviation of Zeeman $g = 2$ factor in theory. In experiments, the in-plane orbital effect can't be eliminated, however the effect is so weak that it hardly affects the analysis of spin Zeeman g factor. Note that our measured the spin Zeeman g factor of around 2 is in good agreement with those by Liu, X. et al. [Nature 583, 221-225 (2020)], Cao, Y. et al. [Nature 583, 215-220 (2020)], and Shen, C. et al. [Nat. Phys. 16, 520-525 (2020)].

Reviewer #2 (Remarks to the Author):

The manuscript by L. Liu et al. reports the observation of new correlated insulator states under relatively large displacement fields (D) in AB-BA stacked twisted double bilayer graphene (TDBG). The authors attribute the observations to the competition of isospin polarization between spin and valley. They also show the measured Landau Fan diagram around the new correlated insulator states. Overall, I think the manuscript would be of great interest to researchers in the fields of graphene-based moiré superlattices and also other moiré systems. It demonstrates a feasible way to realize isospin control in TDBG and could enrich the current understanding of the TDBG system. I would be happy to recommend the paper's publication in Nature Communications if the authors could address the following comments.

Response: We thank the reviewer for the very careful reading of our manuscript and constructive suggestions. We are delighted that the reviewer found our manuscript is of “**great interest**” in the field of twistrionics, which demonstrates a “**feasible way on isospin control**” and “**enriches the current understanding**” of the TDBG. We appreciate the reviewers’ for recommending the publication in Nature Communications.

1. Is the valley polarized correlated insulator states under relatively large D unique for the AB-BA stacked TDBG? Results from an AB-AB stacked device measured under similar D range (about -1.25 V/nm to 1.25 V/nm at $\nu = \pm 2$) would be very helpful.

Response: We thank the reviewer for the constructive comment. Indeed, we have observed valley polarized correlated insulators at AB-AB stacked TDBG, though the quality is not as good as those in the two devices presented in the main text. We fabricate an AB-AB stacked TDBG device using the dual gate architecture with heavy doped Si back gate and Ti/Au top gate. Limited by the gate voltage, we only observe a small part of the valley polarized insulator at $\nu = -2$ with $D = -0.6$ to -0.75 V/nm and $B_{\perp} = 9$ T, as shown in Fig. R4b. This valley polarized insulator at $\nu = -2$ is better revealed in the Landau fan diagram at large displacement field $D = -0.71$ V/nm, which shows the similar features with AB-BA stacked TDBG.

In the revised manuscript, we have included the results of the AB-AB device in the main text and added the Fig. R4 as Fig. S11 in the supplementary information.

Fig. R4. Valley polarized states in ABAB-stacked TDBG. (a), (b) Longitudinal resistance R_{xx} as a function of filling factor ν and displacement field D at $B = 0T$ and $B_\perp = 9T$, respectively. (c) Landau fan diagram at $D = -0.71V/nm$. (d) schematic of Landau levels and the Chern insulator shown in (c).

2. In Fig. 2(d), 2(e), and 3(a), it seems there are abrupt changes of measured resistance (oblique lines on these contour plots) in the electron side under positive D . Are they due to bad contacts?

Response: We thank the reviewer for the comment. Yes, those oblique lines are induced by bad contacts. This is due to the fact that contact fingers are not covered by the top gate. Hence, these areas will become a bad contact when the doping level tuned by back gate reaches the full filling of a moiré flat band.

3. For the g factors obtained in Fig. 1(g) and 2(g), are they symmetric with D ? It is better to show results at both positive D and negative D .

Response: We thank the reviewer for the comment. In principle, the g factors are symmetric with D . However, the g factors can be changed if the dielectric environment is different in experiments due to screening effect. The top and bottom dielectric layer are different. In the case of device D1, the

thickness of the top dielectric layer is $\sim 21.3\text{nm}$ and the top gate is Ti/Au. While the thickness of the bottom dielectric layer is $\sim 16.8\text{nm}$ and the bottom gate is graphite. In fact, we observed the asymmetry behavior at opposite D . As shown in Fig. R5, we plotted R_{xx} versus T curves of $\nu = 2$ valley polarized insulator under the perpendicular magnetic field and calculated the g factors at opposite D . $g \sim 3.54$ at $D = +0.95\text{V/nm}$ is smaller than $g \sim 9.61$ at $D = -0.95\text{V/nm}$. The asymmetry result is due to the external environment and in line with the report by Yankowitz, M. et al. [Science 363, 1059-1064 (2019)].

Fig. R5. g factors at opposite displacement fields. (a), (b) Longitudinal resistance R_{xx} versus Temperature T of $\nu = 2$ insulators at $D = -0.95\text{V/nm}$ (a) and 0.95V/nm (b), respectively. (c) Thermal activation gaps versus B_{\perp} of $\nu = 2$ insulators.

4. The authors attribute the disappear of spin polarized insulator at $\nu=2$ with increasing B_{\perp} to the competition of isospin polarization between spin and valley polarizations. Actually, this observation may also relate to the B_{\perp} induced band reconstruction. To clarify it, the authors should show similar analysis as Fig. 2(g) also for the valley polarized insulator at $\nu=2$.

Response: We thank the reviewer for the comment. We have measured the behavior of $\nu = 2$ valley polarized insulator under the tilted magnetic field in device D2, as shown in Fig. S6. The increase of parallel component indeed suppresses the gap of the $\nu = 2$ valley polarized insulator. To better answer the question, we carry out the same analysis as Fig. 2(g) in device D1. As shown in Fig. R6, we measured the longitudinal resistance of $\nu = 2$ valley polarized insulator under the tilted magnetic field. The perpendicular component is fixed at 6T and the total magnetic field increases from 6T to 9T. The resistance peak at $\nu = 2$ are gradually suppressed and the thermal activation gap are gradually decreased accordingly. The spin Zeeman g factor g_s equal to -1.28 by linear fitting, which further verifies the competition between spin and valley polarizations.

In the revised manuscript, we have included the results of device D1 in the main text and added the Fig. R6 as Fig. S7 in the supplementary information.

Fig. R6. Competition between spin and valley polarization at $\nu = 2$. (a) Longitudinal resistance R_{xx} versus filling factor ν under the tilted magnetic field. Here the perpendicular component is fixed at 6T and B_{total} increases from 6T to 9T. (b) Longitudinal resistance R_{xx} versus Temperature T of $\nu = 2$ insulators shown in (a). (c) Thermal activation gaps versus B_{total} for the $\nu = 2$ insulators.

5. The temperature dependence shown in the inset of Fig. 2(g) need to be further discussed. Is this unusual temperature dependence related to the spontaneously symmetry broken? BTW, in line 130, the 'R(T)' should be 'Rxx'.

Response: We thank the reviewer for the comment. The temperature dependence of the R_{xx} in the inset of Fig.2g is meant to reveal the competition between spin polarization and valley polarization. The unusual temperature dependence includes two different regimes, i.e. B_{\parallel} sensitive at $T < \sim 10K$ and B_{\parallel} insensitive at $T > \sim 16K$. The strong in-plane magnetic field response at low temperature suggests that a spin polarization is a possible ground state which competes with the other ground state- valley polarization state. The insensitive in-plane magnetic field at a high temperature indicates that spin polarization is over dominated by the valley polarization. The different temperature dependence might be related to a temperature driven spontaneous symmetry breaking. However, we don't have more experimental evidences to support this argument, and thus we are reluctant to discuss more about this in the main text. Note that a similar T dependence is also observed in device D1, as shown in Fig.R6 and Fig.R7.

Fig. R7. (a) Zoom in figure of Fig. R6(b). (b) Inset figure of Fig. 2(g).

We have corrected the “ $R(T)$ ” into “ R_{xx} ” in the revised manuscript.

6. The authors claim their TDBG devices are ultra-clean. However, from the measured Landau fan diagrams, the data quality seems not very high. For example, the onset of quantum oscillations is about 2-4 T, which is lower than previous studies, e.g. $\sim 0.5-1$ T in X. Liu et al. Nature 583 221 (2020). Also, the insulating state shown in Fig 3(a) between $\nu=1.0$ to 2.0 looks very broad, which is likely due to some kinds of disorder effects.

Response: We thank the reviewer for the comment. The statement of high quality of the two TDBG devices is mainly based three factors. The first one is the clean interface of the twisted layers, which is free of bubbles in tens of micrometer size with a good twist angle homogeneity. The second one is the low resistance in the metallic regime, from which we obtain a high mobility from Drude model $\rho = 1/ne\mu$.

The onset magnetic field of the quantum oscillations is also a good indicator to quantify the quality of the device. However, the onset magnetic field is also dependent on the band flatness, i.e. fermi velocity, which makes the comparison of the quantum oscillations between different twisted samples more complicated. TDBG is an electric field tunable moiré system, and its band structure can be adjusted from dispersive band to flat band with the increase of the electric field. In the main text, we present the Landau fan diagram at the large electric field, and in this case the Coulomb interactions dominate the transport behaviors where bands are relatively flatter and thus Fermi velocity as well as the Landau quantization gaps are smaller. If we pay attention to the quantum oscillation in dispersive band region, as shown in Fig. R7, the Landau fan diagram at $D = 0$ for device D1, the onset magnetic field of quantum oscillation is ~ 0.75 T.

The broad peak feature in Fig 3(a) between $\nu = 1$ to 2 is due to the mixing of valley polarized insulating

state and Brown-Zak oscillations.

In the revised manuscript, we have added the Fig. R8 as Fig. S14.

Fig. R8. Landau fan diagram of device D1 at $D = 0\text{V/nm}$.

7.A minor comment. The terminologies used in the introduction part need to be further polished. For example, it is more accurate to use 'graphene-based moiré system' instead of 'twisted moiré system' in the 1st to 3rd paragraph, since nowadays there are other moiré systems beyond graphene, e.g. TMDC moiré systems. And some sentences in the manuscript are only correct for the 'graphene-based moiré system', such as line 40-41 on page 2, “Each conduction and valence moiré band has four isospin flavors, involving spin and valley degree of freedoms”.

Response: We thank the reviewer for the comment and we apologize for the mistakes. In the revised manuscript, we use “graphene-based moire systems” accordingly to avoid the confusions.

REVIEWERS' COMMENTS

Reviewer #1 (Remarks to the Author):

Based on the comments and revision provided the authors, I can recommend the publication in Nature Communications.

Reviewer #2 (Remarks to the Author):

The authors have addressed all my comments. I am happy to recommend the manuscript's publication in Nature Communications.

We thank the reviewers for helpful suggestions and the final decision to publish our paper in Nature Communications.